# The Dispute around Same-Sex Marriage in Costa Rica: Arguments and Actions of Conservative Religious Activism (2017–2021)

Arantxa León-Carvajal [1,*] and Andrey Pineda-Sancho [2]

1   Department of Psychology, University of Costa Rica, San José 11501, Costa Rica
2   Center for Research on Culture and Development, Distance State University, San José 11503, Costa Rica; apineda@uned.ac.cr
*   Correspondence: arantxalcarvajal@hotmail.com

**Abstract:** This article examines the role assumed by Costa Rican conservative religious activism during the public controversy that arose in the Central American country regarding the legalization of same-sex marriage. This public debate found its highest point during the presidential elections of 2018. The article is divided into two parts. Firstly, it identifies the arguments used by conservative actors during the controversy, with special emphasis on the conceptions of democracy and human rights that informed their political praxis. Secondly, it ponders the significance that the politicization of religion in a conservative direction has had on the dynamics and democratic institutions of contemporary Costa Rica.

**Keywords:** same-sex marriage; religious politicization; religious conservatism; democracy; human rights; Costa Rica

## 1. Introduction

In 2018, Costa Rica experienced one of the most atypical presidential and legislative elections in its history. After being animated in the beginning by the usual citizen concerns about the state of the economy (with topics such as poverty, employment, and cost of living) and state management (efficiency, corruption), a few weeks before its celebration, an unexpected piece of news significantly altered the process' trajectory and led other concerns to determine its course.

On 9 January 2018, the Costa Rican government and media made the Advisory Opinion (OC 24/17) of the Inter-American Court of Human Rights (IACHR) public. In it, the international organization urged the Costa Rican State, as all member states of the Inter-American System of Human Rights, to guarantee the recognition and protection of the bonds established by persons of the same sex[1]. This call was in line with the protections that the country's legal system guarantees to heterosexual couples and with achieving basic equality for them. In practical terms, this implied moving towards the universalization of all the figures (constitutional, legal, etc.) that protected the relationships of heterosexual couples (civil marriage, common-law unions, divorce, etc.) at the time and thus eliminating any discrimination based on sex or sexual orientation.

As was to be expected, while some sectors of the population welcomed the court's recommendation, others rejected it bluntly. On the one hand, there were groups that had been promoting such claims for several decades, such as those within the LGTBI movements. On the other hand, there were social actors who in recent years had positioned themselves against the legal and socio-cultural changes associated with the emergence and contemporary empowerment of dissident identities from the moral and sexual status quo. This sector was mainly composed of conservative religious agencies and the political-electoral groups closest to them.



Thus, a discussion that had been paralyzed in the deliberative spheres enabled by the Costa Rican political regime (but not in the public sphere) was suddenly reactivated by the IACHR Advisory Opinion and, in the act, took over the electoral dynamics and decided the outcome of the race. As of 9 January[2], the political debate and social conversation tended to focus on the international court rulings. Voting intentions began to favor those candidates who were best positioned at the juncture, and polarization, initially of uncertain significance, advanced to constitute an electoral cleavage around axiological issues of proportions never seen before in Costa Rica.

In the midst of this climate of increasing and effervescent polarization, it was the evangelical candidate, Fabricio Alvarado, of the Partido Restauración Nacional (PRN), who best knew how to channel the anxieties and displeasure that the imminent legalization of same-sex marriage had awakened among part of the electorate. Despite belonging to a non-traditional political organization with a short trajectory in the political-electoral field and an unremarkable electoral record, Alvarado articulated a performativity that allowed him to take advantage of the situation to the benefit of his political project. After a prompt and incendiary reaction to the IACHR Advisory Opinion, and taking advantage of his experience in the field of mass communication, the candidate went from being in the polls' margin of error to "winning" the first electoral round held on February 4 with 25% of the total valid votes.

The evangelical candidate was not only favored by the social friction provoked by the Consultative Opinion, but also contributed the most to prolong and exacerbate it within the electoral framework. Although there were other candidates who tried to obtain benefits from the conservative cause, they did not have the credibility or capacity to gain the support of the sectors of the citizenry and the civil society actors that have historically been part of Costa Rica's conservative activism (the Catholic Church's hierarchy, churches, and evangelical federations). It was Fabricio Alvarado Muñoz, with his rhetorical skills and his background in the defense of "traditional values" (he was then deputy of the republic), who had the best chances to obtain both types of support in the long run.

Although this rise of the conservatives was successfully countered during the electoral process by a heterogeneous conglomerate of socio-political forces that felt compelled to resist for a number of reasons, the political project advocated by Fabricio Alvarado and Restauración Nacional had important repercussions on the country's political dynamics. Not only did it show the potential political power that the religious factor can have at certain junctures, but, at the same time, it evidenced the existence of social struggles around the ways of conceiving and practicing cultural coexistence, politics and democracy, the role of religion in social life, and even around the way of imagining the "defining" features of the national community and those who are part of it. Each of these struggles reflects the diversity of axiological, political, and existential positions that stand out today within Costa Rican society, as well as the obstacles it must overcome to avoid the dissolution of the conditions that make communal life possible.

This article attempts to identify the arguments used by Costa Rican conservative religious activism in the context of the controversy described in the preceding paragraphs as well as the political actions that derived from them. In doing so, it lays emphasis on the notions about democracy and human rights that informed the political praxis of conservative religious groups, analyzing the religious or secular character that they imbued their arguments and actions with during the months that elapsed between the disclosure of OC 24/17, in January 2018, and equal marriage's entry into force, in May 2020.

Based on this analysis, the article assesses the significance that the politicization of conservative religious actors had on Costa Rica's political dynamics and democratic institutions and considers the restrictions that these imposed—in the form of rules or objective constraints—on politicized religious actors. The article's ultimate concern is to determine the impact that religious politicization (whether it involves religious actors in civil society or in political society) has had on Costa Rica's political and democratic culture in recent times.

To achieve these purposes, four empirical sources were analyzed: (1) press articles (informative and opinion) published by media linked to conservative religious activism; (2) press articles published by secular media; (3) press releases written and published by the politicized religious actors themselves during the juncture of interest; and (4) in-depth interviews conducted with actors involved in the controversy. All this information was compiled and organized within the framework of a CLACSO grant entitled "Threats and Challenges to Democracies in Latin America and the Caribbean: Rights in Question?"

The paper is divided into three sections. The first section analyzes the arguments used by conservative religious activism during the controversy, focusing specifically on the conceptions of democracy and human rights present in them. The second section assesses the scope of the "values cleavage" triggered by the IACHR Advisory Opinion and, concomitantly, the mobilizing power of religion and religious (or politico-religious) actors during the situation of interest. The last section presents the conclusion with a critical synthesis of what has been discussed throughout the paper and an evaluation of the impact that religious politicization has had on Costa Rican political culture.

In this article, the notion of "conservative religious activism" is used to include all religious actors or those with religious ties that in one way or another assumed, or wished to assume, an active role in the political arena, whether in the public sphere, as members of civil society organizations and participants in discussions of common interest, or in the electoral field, as members or founders of political parties[3]. The conservative character of such activism derives from its interest in preserving a moral order that is considered threatened by contemporary axiological transformations and, in particular, by the growing influence of the legal and cultural demands raised by feminism and sexual diversity movements (Vaggione 2009). In the Costa Rican case, this activism has been led mainly by the hierarchy of the Catholic Church (through the Episcopal Conference), by the Federación Alianza Evangélica de Costa Rica (FAEC), and by political parties of evangelical orientation such as Renovación Costarricense and the aforementioned PRN.

## 2. Results

This section may be divided into subheadings. It should provide a concise and precise description of the experimental results, their interpretation, and the experimental conclusions that can be drawn.

### 2.1. Arguments and Actions Deployed by Conservative Religious Activism

2.1.1. The Dispute over the Definition of Democracy

As a general tendency, the conservative religious groups that participated in the controversy surrounding the implementation of egalitarian marriage in Costa Rica remained faithful, both at the discursive and practical levels, to the grammar of the democratic game. Although at the discursive level they questioned the legitimacy of certain aspects of the Costa Rican political and legal systems, they usually channeled such complaints through institutional procedures. On a practical level, such a tendency towards democratic procedures was translated into a genuine or resigned acceptance of the fate that finally befell each of the actions that these sectors implemented in order to stop or reverse the legal recognition of same-sex unions in the country.

To a large extent, this way of proceeding largely responded to the constraints of Costa Rican political culture, which defines democracy as a characteristic and mythologized feature of the national being and thus imposes certain limits on political actors' actions (Álvarez 2010). However, above all, it was marked by the political situation in which the dispute took place. The fact that the discussions over same-sex marriage were rekindled[4] in the country in the middle of the electoral campaign, just a few weeks before the presidential elections, meant that the struggle was fundamentally framed in that scenario.

The advisory opinion[5], defended as binding by the Costa Rican government[6], generated a great deal of social commotion in the country and was immediately imbued with the electoral dynamics that were then underway. In this context, conservative religious activism

with political-electoral representation[7] quickly managed to capitalize on and strengthen the social reactions of rejection that arose around the IACHR recommendation and, with it, was able to escalate vertiginously in the voting intentions of the electorate. Thus, an organization of evangelical orientation with 13 years of electoral trajectory and a rather minority presence in Congress, as was the Partido Restauración Nacional (PRN), became one of the unexpected protagonists of the electoral contest and the most voted party in the first round (Pineda-Sancho 2019; Zúñiga 2019)[8].

Although the PNR was not the only political grouping that subscribed to a conservative agenda in moral[9] matters at that time, it was the one that best knew how to take advantage of the situation started off by Advisory Opinion 24/17. The communication skills of its presidential candidate, the journalist and psalmist Fabricio Alvarado Muñoz, who was also then a sitting deputy of the Republic, and, more specifically, the high-flown reaction he had to the recommendation issued by the IACHR Court, worked in favor of the PRN. As soon as the CO was made known and presented as a binding resolution to the Costa Rican public opinion, Alvarado Muñoz denounced it as an act contrary to the majority will of the Costa Rican people and as a flagrant violation of national sovereignty in his social networks (Alvarado 2018a, 2018b). According to his way of understanding and politically constructing the fact, OC 24/17 was the result of a plot (or "compadre hablado", in his own terms) between the Costa Rican government of the time and the IACHR, whose objective consisted of advancing the gender agenda (understood in a progressive key, of course) that both instances had in common at that time through undemocratic means (Alvarado 2018a, 2018b)[10].

With regard to the arguments put forward by Alvarado during the dispute, it is worth noting that the conservative religious agencies had already used a literal, if not extreme, version of the majority principle, generally constitutive of democratic regimes (Bobbio 2005; Lijphart 2008), as the flagship of their cause in favor of the preservation of the Judeo-Christian moral order. In their opposition to the sexual and reproductive rights projects that have been promoted in the country throughout this millennium, these agencies have usually used the majority principle to invalidate all initiatives. For them, the "majority rule", in its simplest application, is the criterion for judging the legitimacy contained in political decisions and the quintessential measure of democratic representativeness. By defending their moral positions, for example in defense of the family and traditional marriage, since their early beginnings, these groups portrayed themselves as representatives of a majority feeling among the Costa Rican population and not as bearers of a specific moral of a specific sector of society. Based on this standing, they argued against all those political initiatives that sought to reform the status quo supposedly protected by the majority of the population and relegated their promoters to the category of threatening agents of the order desired by the Costa Rican people. Under the pressure from this fictitious majority, more rhetorical than empirical[11], they rejected the demands of sexually diverse persons, the bills that were once submitted with the purpose of recognizing the family relationship between persons of the same sex (civil unions, cohabitation societies, etc.), and, of course, the Advisory Opinion 24/17 of the IACHR.

In its most extreme version, the majority principle, defined as a measure of democratic representativeness and legitimacy, leaves virtually no room for minority rights as an element to be considered within socio-political dynamics. In fact, according to the position of conservative religious organizations, governmental decisions (executive, legislative, and judicial) favorable to sexual minorities should usually be interpreted as anti-democratic affronts to the feelings of the majorities and not as legitimate protections within the framework of a pluralistic liberal democracy in which the (social) rule of law supposedly prevails. This way of thinking about democracy has been assumed and extended by the evangelical-oriented parties that have been represented in Congress since 1998[12]. Within civil society, it has also been defended by the leadership of the Catholic Church. This conceptualization was used in discussions related to egalitarian marriage, as is well evidenced in this article, and has been present in other disputes of similar historical and social significance (decrimi-

nalization of abortion; in vitro fertilization; public education on sexuality and affectivity) with the same force.

Thus, it is possible to affirm that Fabricio Alvarado's reaction to the IACHR advisory opinion simply reflected a conception of democracy that already had a long trajectory within the political actions of Costa Rican conservative activism in their struggle to preserve the family institutions traditionally recognized by national legislation. This resource enabled both Alvarado and his party, as well as the religious organizations that gravitated around him[13], to argue against same-sex unions not in a religious key, invoking biblical verses or theological principles, but in an essentially secular line, based on a democratic principle.

The opinion of the IACHR was strongly contested precisely because, according to the criteria of conservative religious agencies, it transgressed the majority rule and particularly because it sought to interrupt the sovereign people's right to decide. In this case, the notion of popular sovereignty was used to denounce the alteration of the internal dynamics of political discussion and decision-making. For its part, the notion of national sovereignty was used to repudiate the alleged annulment of the independent and autonomous character of the Costa Rican nation before international organizations.

With this argumentation, the conservative activists positioned themselves as defenders of Costa Rican democracy and represented their political adversaries as agents willing to ignore the majority's will and national sovereignty to see their ideas reflected in the national legal system. Basically, the organizations that are part of the conservative activism reacted to an action they considered unfair, as did the thousands of people who finally supported the PRN at the polls[14]. It is worth noting that far from being a simple political strategy to limit the recognition of historically relegated rights, the actions of conservative activism are based on a genuine sense of injustice. A sense based on a certain value or axiological framework, and, concomitantly, on a specific emotionality. A feeling of living in a socio-cultural environment "hijacked" by groups contrary to the natural moral order and to democracy itself is what fuels conservative political mobilization. Hence the potency of the narrative articulated by conservative sectors during the 2018 electoral campaign.

It is also interesting to note that the discomfort with the role that international organizations (such as the UN or the IACHR) have or may have in their country's internal dynamics is not exclusive to Costa Rica's conservative sectors but is a widespread feeling among a large part of international conservative activism (Paternotte and Kuhar 2017; Forti 2021; Stefanoni 2021; Graff and Korolczuk 2022). In his talks and writings, a conservative ideologist, Agustín Laje[15], usually warns that today political power is increasingly being absorbed by supranational organizations and that this represents a frank undermining of both the States' sovereignty and democracy. For him, this is the result of the spread of a globalist and elitist agenda that seeks to erode the autonomy of nations and impose a new political and cultural order throughout the world[16] (Laje 2021, 2022).

According to this widespread worldview, international elites would have aligned with local groups to sponsor a general culture transformation, including conceptions of sexuality, sex, gender, and affective bonds. By extension, they would dismantle the traditional meanings and practices of these spheres of existence, especially those rooted in the Judeo-Christian tradition[17]. Within this framework, the defense of Christianity and institutions such as the nuclear family and heterosexual marriage would be a kind of popular resistance to the onslaught of the globalist elites.

During the controversy discussed here, the conservative religious organizations, agglutinated around the figure of Fabricio Alvarado, took great pains to present the IACHR, the Organization of American States, the United Nations (UN), and political parties such as the Frente Amplio, Liberación Nacional, Unidad Social Cristiana, and the ruling Partido Acción Ciudadana[18] as vehicles of this "globalist ideology" in Costa Rica (Citizen Action Party). This last group was finally pointed out as the most visible face of that supposed global trend at the domestic level (Alvarado 2018a, 2018b; Radio Monumental 2018). According to this self-serving interpretation of the facts, the aforementioned actors undermined the

internal instances of democratic deliberation and ultimately affronted the popular will through their actions or omissions.

In such an electoral context, the narrative articulated by Fabricio Alvarado proved adequate to capture the attention of the most conservative sectors of the citizenry (Pignataro and Treminio 2019). After being very close to the polls' margin of error during most of the campaign, the evangelical candidate managed to sneak into the second electoral round with 25% of the total votes and to win the favor of 859,319 people (equivalent to 39.41% of the valid votes cast in that round) in the runoff. However, such a performance was not enough to win the presidency. During the second round, his political project was defeated by a candidate who had positioned himself as a defender of human rights and as the "absolute reverse" of the ideals represented by the National Restoration Party, namely, the pro-government Carlos Alvarado[19] (Cortés 2019; Pineda-Sancho 2019; Pignataro 2021).

After being defeated at the polls, the existence of a moral majority, of which Fabricio Alvarado claimed to be a representative, was put in doubt[20]. However, this did not prevent the former candidate for Restauración Nacional, nor his allies and representatives in Congress, from continuing to invoke the same principles to oppose the imminent entry into force of equal marriage. From the Legislative Assembly's own tribune, the written press, and social networks, the conservative religious organizations continued denouncing the IACHR's Advisory Opinion as an outrage to popular and national sovereignty and as an anti-democratic imposition. Not even a sentence of the Constitutional House of Costa Rica, which ratified the binding nature of the Advisory Opinion and gave Congress 18 months to align the national legislation in line with it[21], was able to change the opinion of the sectors that supported the conservative cause during the campaign. For many of them, the decision of the constitutional court, the highest body to construe the Magna Carta in force in Costa Rica, simply bent to the dictates of the IACHR and to the interests of sexual minorities through a complacent resolution.

Based on this conviction, the conservative positions represented in Congress proposed projects to create alternative legal figures to same-sex marriage (e.g., civil union or cohabitation[22]). This path proved to be politically useless since they did not even have majority support from the other legislative benches. Then, they implemented actions to delay same-sex marriage's entry into force for a few more months. In the long run, this effort did not prove fruitful to their intentions either[23]. After these failures, conservative religious activism lost even more strength and legitimacy. By that time, it was confronted not only with the resolution of their government and the IACHR but also with the Constitutional Chamber criteria and the dominant position in the Legislative Assembly, a body that these sectors had always considered the most important sphere of democratic representation in the socio-political framework.

Finally, in May 2020, equal marriage became a reality in the country, and the conservative sectors could do nothing to prevent it. After having presented themselves as legitimate representatives of the majorities, they succumbed to a rather complex democratic dynamic, in which "majority rule" is neither the source of legitimacy nor the only criterion for making decisions with collective repercussions[24]. Faced with such a reality, the last resort of conservative activism consisted of relying on instruments for the protection of minorities in order to remain strong before the "disarticulation" of the moral order they had so well protected. Ironically, conscientious objection—a figure essentially designed to repel or avoid the potential tyranny of majorities over the lives of individuals (Gallardo 2021) and, therefore, typically associated with the modern paradigm on human rights—appeared on the political scene as the last possibility of resistance for Costa Rican conservatism. In this way, it was the tools usually wielded by their political adversaries that allowed them to articulate a final strategy to defend their cause.

In the months immediately following equal marriage's entry into force, conservative activism with representation in the Legislative Assembly promoted five bills[25] to make conscientious objection a right recognized and protected by the Costa Rican State. According to statements made by former presidential candidate Fabricio Alvarado in May 2020, on

the same day equal marriage was adopted in the country, the approval of conscientious objection as a permanent prerogative of citizenship would at least allow conservative sectors of the Costa Rican population "not to be victims of persecution for standing firm on their principles and not be forced to do anything they do not agree with" (Alvarado 2020a); This would include the possibility of refusing to celebrate marriages between persons of the same sex (in the particular case of Republic judges) and the right to publicly express their disapproval of such unions and of sexual diversity in general.

Although only one of the five initiatives was approved by Congress[26], their very existence reflects the tenacity with which conservative activists defend their existential positions. As Mariela Puga and Juan Vaggione point out, resources such as conscientious objection have started to be used by conservatism throughout the region as a strategy to counteract the actual or potential advancement of sexual and reproductive rights. Where such rights have been recognized and incorporated into the legal system, conscientious objection has emerged as the privileged way to make their effective application operationally unfeasible (Puga and Vaggione 2018). This strategy has the advantage of being presented not as a religious prerogative but as a continuation of the human rights' paradigm, usually invoked by those who strive for the expansion of sexual and reproductive autonomy.

Hence, conservative activism went from relying on the strength of majorities to taking refuge in the rights of minorities. In addition to reflecting a notable change in argumentation and political stance, this paradoxical use of conscientious objection confirms that religious agencies that move in the political arena and seek to influence public opinion increasingly use secular arguments to defend their causes.

As it has been well and clearly stated by Monte and Vaggione (2019, pp. 111–12), these agencies tend to channel their opposition to "feminist and LGBTI demands through the discourse of rights and the intervention in legal-state institutions such as legislatures, Congress and judicial processes". This could be seen as a plain political strategy aimed at widening the fronts of struggle. Moreover, it would be a course of action that would have the value of not giving away the exclusivity of any space to their political adversaries (hence its strategic character), but that can also be read as the result of the genuine assimilation of a political language that is not constructed from the religious and that prevails both in the political-electoral field and in the public sphere. This would mean that today's politicized religious groups do not define the mood of the political landscape but adapt to it in a rather reactive and, to a certain extent, resigned way. In the first case, we would be dealing with a sort of planned "doublethink", to use George Orwell's expression, while in the second case, we would be dealing with the inertial "worldliness" of the original religious positions.

### 2.1.2. The Dispute over the Definition of Human Rights

Nothing reflects the aforementioned transformation better than the space that conservative religious organizations (whether evangelical or Catholic) grant to the notion of "human rights" in their current thinking and discourse. Instead of rejecting it ad portas, as they did when the modern paradigm on human rights began to take shape (Ruiz 2016), during the last few years, they started considering themselves as precursors, defenders, and promoters of this current. However, in doing so, they have refused to accept certain ways of understanding the notion and, instead, have tried to position their own definition with more or less coherence and with a certain degree of systematicity. As in the case of the term "democracy", in this instance they attempted to dispute the meaning and scope of human rights. Through this move, they sought to distance themselves from the stereotypes that represent them as anti-rights sectors and thus fight against progressive activism's "monopoly of legitimate interpretation" of the notion.

In the Costa Rican case, this orientation became evident in the midst of the analyzed controversy but had been generally present during the discussions of bills associated with the acceptance of same-sex unions. In the midst of them, both the Catholic and Evangelical factions that participated in the debates presented themselves as defenders of

human rights and emphatically denied that their opposition to such projects responded to a discriminatory vocation or a logic contrary to human dignity.

During the disputes, the logic followed by these actors (such as the Conferencia Episcopal de Costa Rica, the Federación Alianza Evangélica de Costa Rica, or members of political parties of evangelical inspiration, such as PRN), did not consist in overriding the value of human rights but in questioning whether the legalization of same-sex couples in any of its possible modalities had the status of a human right. As a general rule, it was presented as a sort of "false right" and as a claim that, if approved, would infringe on "authentic rights"[27], such as the right to form a nuclear, heterosexual family and the correlative right to marry a person of a different sex[28].

The strategy of conservative religious activism consisted of stressing that socio-legal figures such as "traditional family" and "marriage", unlike same-sex unions, were expressly recognized as fundamental rights both by the Costa Rican Political Constitution and by the main international human rights instruments. In the latter case, they frequently invoked Article 16 of the Universal Declaration of Human Rights and Article 17 of the American Convention on Human Rights, which recognize "the right of men and women to marry and to start a family", and the family institution as "the natural and fundamental part of society".

Based on these premises, conservative religious sectors reminded political, legal, and governmental agents that they had the obligation to "safeguard the protection of the family based on monogamous marriage and to protect its unity and stability". In addition, they tried to make them understand that supporting initiatives contrary to this obligation was an improper course of action, contrary to human rights. In this way, in terms of the conservative worldview, the projects aimed at universalizing access to marriage in the country were initiatives contrary to already recognized human rights. According to this order of ideas, the progressive sectors that present themselves as defenders par excellence of human rights, usually embodied by feminist and LGTBIQ+ activism, not only distort the meaning of such rights but also make a convenient and self-interested selection of them.

According to these arguments, the progressive sectors that participated in the discussions around the possibility of protecting same-sex unions tended to minimize, relativize, or ignore those rights that did not coincide with their worldview and/or their political-existential interests[29]. They disregarded rights such as those recognized by the aforementioned Articles 16 of the Universal Declaration of Human Rights and 17 of the American Convention on Human Rights and would have attempted to pulverize rights such as freedom of expression, religion, and conscience in the name of an inadequate use of the principles of equality and non-discrimination for their own benefit.

For the actors who were part of conservative religious activism, their oppositional stance simply could not be labeled discriminatory. Firstly, because such regulation was and is not a human right as such; and secondly, because their dissent was supposedly protected by genuine human rights, including the aforementioned freedoms of expression, religion, and conscience. In this way, they repeatedly denounced that their opinion tended to be improperly discredited and that the original meaning of human rights was being adulterated by movements related to the particular interests of sectors such as the sexually diverse population.

This perception of injustice and the feeling of being in the presence of a hijacking of the true meaning of human rights by progressive sectors are illustrated in a book written by Fabricio Alvarado in 2016, a couple of years before the controversy analyzed here, and republished in a digital version in March 2020, just two months before the entry into force of equal marriage in Costa Rica. In this insightful text, entitled "Christians in politics", Alvarado Muñoz points out that:

We live in a generation where the issue of human rights has become a source of discord and a generator of attacks against politicians who dare to have a discourse contrary to the bearers of causes that are contrary to Christian faith and values. Specific groups, such as feminists and the LGTBI movement, have taken over the issue, and when talking about human rights, it seems that they only focus on the "woman's right over her own body"

(which is nothing more than the ideology promoting abortion in all its forms) and on the "right of gays and lesbians to marry their partner" (which is nothing more than the destruction of the historical institutions of marriage and family). In other words, as the years have gone by, these groups have turned their personal desires, first into "social" needs, and now into "rights", and have succeeded in positioning their discourse at the global level (Alvarado 2020b, p. 34).

This fragment of the book contains almost all the arguments discussed in this section. Here, Alvarado states he is aware of the current disputes surrounding the notion of "Human Rights", but he does not identify them as part of an inevitable process of contemporary political dynamics, in which axiological plurality is usually the norm rather than the exception, but as the result of an outrage to the meaning and legitimate scope of human rights. According to this framework, this outrage may have derived directly from the personal desires of those who are part of vindictive movements such as feminism and LGTBIQ+ and from their will to undermine the foundations, both cultural and natural, of traditional social orders.

As they recurrently argue, for both evangelical and Catholic conservative religious activism, it is almost evident that the figure of traditional (heterosexual) marriage enjoys the status of an authentic human right not by virtue of its capacity to respond to a simple whim of particular sectors of society[30]-as would be the case with unions between persons of the same sex—but for the purpose it pursues and the role it allegedly plays in the social fabric and, more specifically, in the future of the human species. Although usually based on theologically inspired principles such as "right reason" and "natural law"[31] (more typical of Catholic theology than evangelical theology, it should be acknowledged), in this clearly utilitarian argument, heterosexual marriage receives its status as an authentic right by virtue of its reproductive purpose and its contribution to the temporal permanence of humanity, understood now as the work of the Creator. According to this worldview, heterosexual marriage is useful and important for societies since it guarantees the continuity of the species, while same-sex unions (and, by extension, homosexuality and lesbianism) would rather go against this need and, therefore, would be detrimental to the common good. Hence, it cannot and should not be considered a right, be it human, fundamental, constitutional, or civil.

This particular argument is by no means new or exclusive to Costa Rican conservative activism. Similar theses have been developed in other parts of the world. In Latin America, they were used in countries such as Mexico, Argentina, and Chile, where the debates around the potential universalization of the marriage figure took place before or concurrently with those in Costa Rica. Here, too, these arguments were defended both by actors with a clear religious background, such as the Catholic Episcopal Conferences and the Evangelical Federations that are present in almost all the countries of the region, and by secular public figures, among whom the Argentinean lawyers Jorge Scala and Nicolás Marqués and the political scientist Agustín Laje are outstanding. As mentioned in the previous section, these actors have been responsible for giving an intellectual orientation and secular nuances to contemporary Latin American conservatism. All three have become secular "apostles" of the conservative cause and nodes of the anti-gender discourse in the subcontinent.

Although it is fallacious, since the existence of same-sex marriage does not prevent heterosexual couples from reproducing, whether inside or outside the institution of marriage, this argument proved nonetheless attractive. Its strength lies not in its logical consistency but in its ability to rally believers and nonbelievers around a driving idea based on a supposed commitment to the common good of humankind. Thus, opposition to egalitarian marriage would not be defined negatively as the simple desire not to recognize a right for certain sectors of society, but first and foremost as a just cause. Here again, the idea of what is fair and what is unfair depends on the axiological framework from which reality is assessed.

### 2.2. Limits and Scopes of Conservative-Religious Power

After the defeat at the polls, the political power of religious conservatism was relativized and, to a certain extent, diminished. Although it had the capacity to mobilize a significant sector of the electorate in favor of its cause, it did not have enough support to gain access to the Republic presidency, as it was thought at some point during the electoral process, or to become the representative, consequently, of the majority of the population. Beyond the vertiginous growth experienced by the PRN in the voting intentions for the first electoral round and its historic "triumph" in such elections, the analysis of the real support it received throughout the 2018 electoral process shows that the political party was far from winning the sympathies of most of the people registered on the electoral roll. During the first round, it barely received the support of 16.4% of the electoral roll, reaching only 26.1% in the runoff.

Moreover, the polarization that distinguished a prolonged stretch of the campaign finally led many sectors of the population to a disapproving, even hostile, attitude towards conservative religious activism and the link between religion and politics. Thus, while the PRN managed to gain unprecedented electoral support within the general trajectory of evangelical-oriented parties (Pineda-Sancho 2019), within the same movement it gained the animosity of other sectors of the citizenry and, with it, compromised its future possibilities of political-electoral growth. The same conjuncture that allowed it to partially abandon its status as a niche party (Kernecker and Wagner 2019) could have foisted a heavy burden on its short- and medium-term political expectations.

These facts show the limits that conservative religious activism faces in making its political project an attractive option for the population as a whole and reveal that within Costa Rican society there is a political and moral heterogeneity that does not offer political groups with restricted or single-thematic agendas a great growth margin. In exceptional situations, such as the one experienced during 2018, these groups may indeed receive special attention from the electorate, but it is unlikely that this will translate into prolonged and massive support from the majority.

This existential diversity and the great dissociation between the citizenry's demands and the political parties' offers that characterize Costa Rican politics play against this possibility. For at least 20 years, political parties have shown themselves to be less and less capable of representing the interests of the population and of securing their long-term loyalty. This has been expressed in an increasingly fragmented party system, high political volatility, and great electoral uncertainty (Alfaro 2020; Alpízar 2021), since no political party has the capacity to build strong and durable hegemonies over time.

In addition, it is important to highlight that, although there have been political parties with agendas focused on the protection of a certain moral order in the country for more than 40 years, the concerns that they convey and try to position in the public sphere have rarely been assumed to be priorities by Costa Rican citizens. These tend to focus their attention on material problems such as employment (unemployment, quality of employment, etc.), insecurity, and political corruption (Rodríguez et al. 2019). Thus, although post-material values are gaining ground among many sectors of society (Treminio and Pignataro 2019; Gómez 2020), they do not yet have the power to generate major political-electoral cleavages, with the obvious exception, of course, of the situation discussed here.

Against this backdrop, the political parties of evangelical orientation have two alternatives. The first one consists of broadening their ideological-programmatic offer with a view to leaving, once and for all, the niche from which they have usually operated within the political-electoral field. This would offer them the possibility of competing with the mainstream parties, already in decline, and of attracting a greater number of possible followers to their orbit (with no guarantee that this will happen, needless to say). The second option is to remain faithful to the monothematic proposal that has distinguished them so far, hoping that a situation similar to that of 2018 will arise again or with the expectation that the moral cleavage will cease to be secondary and become the leading and structuring factor of the political dynamics. Both options represent a risky bet and have

become a real dilemma for the members of these groups. In fact, this became more than evident in the months prior to the second round of the electoral process discussed here, since for the two political parties that remained in dispute, it became impossible, and in one case, even undesirable, to keep the debate strictly limited to the moral topic that marked the first round's results. At the onset of the runoff, the media, the format of the debates, the demands of the electorate itself, and the strategy followed by the rival candidate and his team inexorably widened the boundaries of the contest and took it to a frankly unfavorable terrain for the evangelical party.

During the almost two months that elapsed between one round and the next, both the ideological and programmatic limitations inherent to Restauración Nacional and the poverty of its organizational structure became evident. Its own mistakes, such as the presentation of a brief government plan, its restricted knowledge of national problems, and its manifest incapacity to work in a cohesive manner, were skillfully exploited by its adversaries (Mora 2019). At the same time, they marked the political mood of an important portion of the electorate. Facing the final round, the general image of Fabricio Alvarado was one of a character incapable of governing the country properly[32], that is, with a solid plan and work team and in tune with the complexities of the state's institutionality and the Costa Rican political regime.

Despite the efforts made by the PRN in record time to overcome its structural limitations, inherent to its narrow ideological-programmatic profile and its weak internal organization, nothing this party did was enough to reach the desired goal. When put to the test with a conventional political party with a broader thematic scope and past experience in the state administration, such as the Citizen Action Party, the party represented by Fabricio Alvarado was unable to obtain the majority support of the population[33]. This evidently put an end to its aspirations to reach the presidency and weakened its power to assert its political agenda, now in its role as an opposition force[34].

In addition to these extrinsic and intrinsic limitations, the final result of the controversy shows the influence that the codes and institutions of the Costa Rican political regime have, by way of objective constraints, on the actors that play a part in the political field. As a liberal democracy of a representative type, this framework admits mechanisms for making decisions of collective interest that go beyond the majority rule and, at the same time, contemplates specific devices for the protection of individual rights in the face of potential abuses by civil authorities or even by the majority of the population. This means that in practice, not everything that is decided in the country passes through the sieve of the Legislative Assembly and that some matters of interest are rather resolved through national and/or international law. This was precisely the case with the legal recognition of same-sex couples between 2018 and 2020, when it finally came into force.

These constraints explain, to a large extent, the eminently institutional orientation of the actions implemented by conservative religious activism during the controversy. They also allow us to understand, to a certain extent, the progress made in the recognition of sexual and reproductive health rights (SRHR) in the country. Despite the presence of conservatism in civil society, in political organizations, and in the mindset of an important sector of the Costa Rican population, the claims associated with this type of right have managed to gain ground at the legal, institutional, and cultural levels[35]. The growing trend towards moral pluralism clearly shows a process of socio-cultural transformation that no political actor has been able to stop so far and that will be difficult to reverse (Pineda-Sancho 2022).

Seen from the perspective of the conservative religious sectors present in the political society's perspective, the 2018 elections yielded rather paradoxical results. Although they allowed them to obtain their most significant representation in the Legislative Assembly in history, during their four years of administration they were unable to stop the implementation of equal marriage or advance their moral agenda through laws or concrete public policies. The weight of the courts (IACHR, Constitutional Chamber), the defeat at the polls, and the consequent change in the dominant positions in the Legislative Assembly rendered them feeble to significantly influence the national agenda.

## 3. Discussion

This article stems from an interest in giving an account of the arguments used by Costa Rican conservative religious activism in the midst of the actions that it developed to oppose and resist the legalization of equal marriage in the Central American country during the period 2017–2021. The paper not only identified such arguments but also unveiled the concrete political action that found support in them and was concerned, ultimately, to unravel the possible impacts that these could have on Costa Rica's political culture and institutionality.

The focus of attention concentrated on the significance that the politicization of conservative religious actors had on the electoral dynamics and democratic institutions historically existing in Costa Rica during or as a consequence of the analyzed conjuncture. The paper also addressed the limits that these dynamics and institutions, by way of rules or objective constraints, imposed on politicized religious actors and their political-existential proposal within the framework of the debate developed around the legal recognition of same-sex unions during the 2018 electoral process and afterwards. The starting point of the phenomenon's interpretation consisted in assuming that the conglomerate of actors' proposals introduced new worldviews, concepts, and practices in local political dynamics and, at the same time, to remember that, however new this proposal may look, it does not start from a vacuum but is part of a political, institutional, and legal ecosystem that, in one way or another, guides it along certain paths and marks its limits as well as its possibilities in the short term. This starting point proved to be useful both to understand the balances that the dispute produced and to understand the course of action followed by conservative religious activism throughout the dispute.

That political ecosystem would explain the fact that the controversy was resolved through mechanisms and criteria contemplated by the structure of the Costa Rican political regime, which is essentially democratic and liberal. However, above all, it allows us to understand the reliance that politicized religious actors, whether in civil or political society, usually showed towards that structure. At the level of argumentation, for instance, it was observed that at no time did conservative religious activism challenge democracy as a political regime, while at the level of action, it was found that they generally opted to remain attached to legitimate mechanisms, that is, mechanisms contemplated by the institutions and norms of the Costa Rican political regime and accepted, with greater or lesser degrees of conviction, by the range of political actors present in the country's social space. Although in some moments of the dispute they vehemently questioned the legitimacy of the decisions made by state or governmental actors who, strictly speaking, had the democratic faculties to act in the way they did, they almost always channeled their objections through institutional channels.

This factor would shed light on the secular turn given by conservative religious activism in the context of the dispute. The fact that their arguments have been essentially based not on direct appeals to the word of God or on obvious allusions to religious doctrines, as is usual within fundamentalism, but on appeals to democracy and human rights is revealing of the impact that hegemonic political dynamics have had and continue to have on the political practice of the actors that make up conservative activism. Although the politicization of religion is usually interpreted as a sign of the non-secularization or desecularization of society as a whole and of politics in particular, the case analyzed leads us to conclude that, in its general contours, the Costa Rican political sphere is an increasingly secularized space. For although religious actors, ideas, and symbols are present in it, neither its lingua franca nor its operation logics are dictated by such factors. Instead of defining the rules of the game, politicized religious actors have been trying to assimilate, more out of a need for survival than sheer will, a political language that they do not control and that, in many aspects, does not convince them either. It is a fundamentally secular language from which no political actor can escape and within which the antagonisms between the diverse ideologies and political projects existing in Costa Rica develop.

This political language, which works as a sort of background that is not always transparent to its users, is how we gain insight into, to a certain extent, the fact that the actors converge in the use of certain notions, concepts, and ideals (such as democracy and human rights, etc.), as well as in the dispute to define their ultimate significance[36]. It could be said, then, that the politicized conservative religious actors have genuinely assumed the notions of democracy and human rights that are so important for the political orders of late modernity as their own. In doing so, they have entered into a fierce struggle to establish their definitions. This latter action is typical of modern social dynamics in which, according to Max Weber's famous expression, the "polytheism of values" prevails and the tendency to antagonism is usually as present, if not more so, than in other historical periods.

It is in this last point that the greatest imprint of conservative religious activism in the political dynamics of contemporary Costa Rica can be found. Beyond the strictly religious dimension, which has always been present in the country's political culture, albeit as a distant referent (Pineda-Sancho Forthcoming), what today's religious conservatism politicization left behind were narratives, worldviews, and concepts on democracy and human rights with the capacity to compete before the citizenry with the narratives defended by other political actors and sectors of society, among which stand out progressive activism and sectors more akin to the liberal and reformist political traditions. In terms of narratives, worldviews, and concepts on democracy, for example, the actors that are part of the conservative religious conglomerate have shown themselves to be close, at least in critical situations such as the one analyzed here, to populist-oriented notions on democracy. Despite the fact that it would be out of proportion to identify them as the introducers of the populist tendencies that have blazed a trail in the Costa Rican political culture during the last few years, it is fair to point them out as part of the group of actors that contributed the most to promoting them. Their populist drift was made evident during the dispute over equal marriage and was especially exposed in their arguments about the idea of democracy. These arguments included a conception of the people based on the "us/them" distinction ("us", the people, vs. "them", the globalist elites); a polarized and hyper-electoralist vision of the sovereignty of the people; a questioning of the legitimacy of non-electoral institutions, such as constitutional and international courts; and, finally, a sovereigntist stance on the interpretation of national autonomy. These traits have been identified by authors such as historian and sociologist Pierre Rosanvallon (2020) as characteristic of the populist ethos and the type of democracy with which it feels most comfortable.

As already mentioned, these notions of democracy not only reflect the conflictive nature of current political dynamics (generally channeled through agonistic channels), but also reveal the instability of the concept and the ideals it inspires. Although the word democracy is usually on the lips of all socio-political actors in Costa Rica, the case analyzed here reveals that it does not have the same meaning for all of them. This makes it possible for all actors to speak in the name of democracy and position themselves as its defenders without necessarily implying the elaboration of a lie or self-deception. In the end, what is at stake is a struggle for the "monopoly of the legitimate interpretation" of the concept, something that clearly does not respond only to exegetical interests but above all to competing worldviews, sets of values, and political projects.

This struggle makes it possible to speak of democracy in pursuit of the most dissimilar ends. While conservative activism invokes it with the purpose of preventing the recognition of sexual and reproductive rights claims, progressive activism invokes it in a diametrically opposite sense. For the former, such demands are an affront to the principle of majorities, which should constitute the cornerstone of democracy. On the contrary, for the latter, they represent manifestations consistent with the spirit of guarantees, inclusiveness, and pluralism of any regime that is truly democratic. In this sense, the word democracy is both an object of dispute and a weapon to start disputes around other topics.

Although the power of the conservative religious project is rather limited, as is well shown by the fate of the controversy that has been the subject of this article, it has already left its mark on Costa Rican political culture. On the one hand, it has revived the presence

of religion in the country's public sphere and in the political-electoral field. On the other hand, it has opened a place, albeit of a secondary nature, for disputes over moral issues within the political dynamics at both the electoral and civil society levels. These traces undoubtedly represent challenges for the advancement of SRHR and also guarantee the continuity of conflict around the promotion of such rights. This, in turn, poses challenges of great importance both to the Costa Rican political regime, in its most formal facet, and to the culture that would guarantee sociocultural coexistence and the conditions for recognition among the different political-existential positions that populate the social space.

Such challenges reach an even higher level of complexity and significance when noting that the political practice of religious conservatism today follows secular and not religious paths. The appropriation and use of the notions of democracy and human rights for their own ends reveal the political sophistication that the movement has acquired and, in a correlative manner, demonstrate their adaptive capacities. By acting in this way, they have achieved, with or without intending to, new ways of legitimizing their agenda, have managed to broaden their political repertoire, and, finally, have converged with political trends of greater societal repercussion, such as right-wing populism.

**Author Contributions:** Conceptualization, A.L.-C. and A.P.-S.; methodology, A.L.-C. and A.P.-S.; investigation, A.L.-C. and A.P.-S.; writing—original draft preparation, A.P.-S.; writing—review and editing, A.L.-C. and A.P.-S.; supervision, A.P.-S.; project administration, A.L.-C. and A.P.-S.; funding acquisition, A.L.-C. and A.P.-S. All authors have read and agreed to the published version of the manuscript.

**Funding:** This research was funded by CLACSO in the framework of the call for proposals "Threats and Challenges to Latin American Democracies: Rights in Question?".

**Institutional Review Board Statement:** This research was reviewed according to the guidelines of the ethical Statute of the CLACSO.

**Informed Consent Statement:** Informed consent was obtained from all subjects involved in the study.

**Data Availability Statement:** Data is unavailable due to privacy and ethical restrictions.

**Conflicts of Interest:** The authors declare no conflict of interest.

## Notes

[1] This advisory opinion was expressly requested by the Costa Rican state on 18 May 2016 (IACHR 2017, p. 3). In this case, the Costa Rican government exercised its right to consultation granted by Article 64 of the American Convention on Human Rights in order to clarify whether the San Jose Pact protected the recognition of name changes for individuals based on their self-perceived gender identity and if the American Convention recognized property rights derived from relationships between same-sex couples (Presidencia de la República 2016). According to the promoters of the initiative, the Costa Rican government sought to resolve by this means the historical lack of state recognition (legislative, judicial, etc.) of the demands and rights of sexually diverse citizens (González 2020).

[2] The Inter-American Court of Human Rights adopted the Advisory Opinion discussed here on 24 November 2017 (IACHR 2017), but the Costa Rican government did not receive it until 9 January 2018 (Presidencia de la República 2018), the same day that the international court issued a press release on the matter (IACHR 2018).

[3] The category includes civil society organizations such as the Conferencia Episcopal de Costa Rica (CECOR) and the Federación Alianza Evangélica Costarricense (FAEC), political society actors such as the Renovación Costarricense (PRC) and the Restauración Nacional (PRN), political parties with evangelical orientation, and the Costa Rican Catholic Church as the official church of the Costa Rican state. These were the institutions that were most actively and visibly involved in the controversy discussed in this article. However, it is necessary to highlight that in practice, it was the Partido Restauración Nacional (PRN), led by Fabricio Alvarado Muñoz, the leading conservative actor, that was the most prominent actor in the dispute and the one that was at the forefront of the conservative cause. This happened in such a way by virtue of the electoral nature that the controversy acquired and the ability that the PRN had to put it at the service of its own political ends.

[4] The struggle to ensure the legal recognition of family and affective unions formed by persons of the same sex began to take shape in the country at the beginning of this millennium (Jiménez 2017). From 2003, when the exclusively heterosexual nature of the marriage figure then in force in Costa Rican legislation was first challenged in court, until the beginning of the period studied here (2017), five bills were presented (the last of them in the now distant year of 2012) aimed at establishing a legal framework to protect sexually diverse couples (Maroto 2021). However, the moral conservatism prevailing among most of the members of

the political-electoral field of that period, as well as the particular opposition that these initiatives encountered in the political parties of evangelical orientation, on the one hand, and in the leadership of the Catholic Church, on the other, paralyzed any reform attempt.

5　　It is worth noting that, in Advisory Opinion (OC-24/17), the IACHR not only endorsed the recognition of the rights derived from the family relationship between persons of the same sex, it also positioned itself in favor of the recognition of self-perceived gender identity by all its Member States (IACHR 2017, p. 87). However, this second aspect was relegated to the background during the discussions that took place in the last stretch of the 2018 electoral campaign.

6　　In this case, it is important to note that the consultation before the IACHR Court was conducted by the Costa Rican State itself in May 2016 (Presidencia de la República 2016). It was the government headed by Luis Guillermo Rivera, of the Partido Acción Ciudadana (PAC), which requested the opinion of the Court on the two issues mentioned above (rights derived from the family relationship between persons of the same sex and the right to self-perceived gender identity) (IACHR 2017, p. 3) and which received, shortly before the end of his four-year term as head of the Costa Rican State, Advisory Opinion 24/17 in its entirety.

7　　Conservative religious activism with political-electoral representation includes, in a restricted sense, political groups with an explicitly religious orientation that have direct participation in the political-electoral field (political society) and presence in the Congress of the Republic. For the case discussed in this article, specific reference is made to the evangelical parties Renovación Costarricense (PRC) and Restauración Nacional (PRN).

8　　The opposition to same-sex marriage was not the exclusive stance of religious parties with representation in Congress. This position was also taken, during the election campaign, by members of non-religious (or, at least, not directly religious) political parties and by secular actors in civil society. If the paper doesn't give enough space to those actors, it is because their role in the controversy was not particularly prominent and because the political project of these actors is not primarily based on religious concerns and ideals.

9　　In addition to PRN, the Partido Renovación Costarricense (PRC) was also at power at that time, with representation in the Legislative Assembly. This party introduced the conservative evangelical imprint in Congress at the end of the 20th century and was the most important evangelical party in the Costa Rican political scene during the first two decades of the 21st (Pineda-Sancho 2019; Zúñiga 2019).

10　　Although it is true that Fabricio Alvarado's reaction benefited from the long conservative trajectory around the defense of the "traditional moral order", a good part of the success achieved responded to the existential anxieties that the PRN candidate was able to awaken among the Costa Rican population through his narrative and performance. With his prompt denunciation of plots, his defense of majority democracy, his defense of national sovereignty, and his belligerent denigration of policies favorable to the recognition of sexual diversity, Alvarado activated a moral panic with great electoral repercussions among a sector of Costa Rican society. In a very short time, the candidate defined the contours of a threat to the traditional values of Costa Ricans, managed to have this perception of a threat replicated by some media, and was quickly assumed to be their own by some sectors of the population.

11　　For several decades, the preponderantly conservative nature of the political parties and the Legislative Assembly seemed to support this "majoritarian" position because, as mentioned above, the conservative predominance in important decision-making bodies like Congress prevented any alteration of the sexual order naturalized by certain versions of religion and by the Costa Rican legal system (including the Constitution) (Maroto 2021). It is evident that, feeling this safeguard, conservative religious activism saw in the majority principle a reliable criterion to cement its political positions and arguments. Despite the growing presence of feminist and sexually diverse movements in the Costa Rican public sphere, conservative agencies failed to foresee early enough that sooner rather than later this foundation would lose its bearings.

12　　In Costa Rica, there have been evangelical political parties since 1981. However, it was not until 1998 that they managed to obtain representation in the Congress of the Republic. Since then, these groups have been able to maintain a minority presence in Congress and a limited share of power within the electoral political field. In both realms, they have been champions of the conservative cause in favor of maintaining the moral status quo in Costa Rica (Pineda-Sancho 2019, Forthcoming).

13　　As already pointed out, at the beginning of 2018, the PRN led by Fabricio Alvarado was not the only religiously rooted actor mobilized around the conservative moral agenda, nor was it the only one to react negatively to the IACHR Court's OC 24/17. Although they played a less prominent role, organizations such as the Federación Alianza Evangélica Costarricense (FAEC) and the Conferencia Episcopal de Costa Rica (CECOR) also contributed to stirring up the controversy. In general, they tended to favor the PRN's electoral option. In that sense, the joint communiqué that the two organizations published on January 18, 2018, it is illustrative since in it they not only launched a message of rejection of egalitarian marriage but also made a call "to all Christians and all citizens to participate in the February 4 elections, meditating before God and their consciences their vote" (CECOR and FAEC 2018).

14　　According to a public opinion survey conducted by the Center for Research and Political Studies of the University of Costa Rica between 15 and 17 January 2018, one week after the announcement of the Advisory Opinion, 75% of the Costa Rican population was aware of the resolution at that time, and among this percentage, 59% of people expressed opposition to its contents (CIEP 2018a, p. 8). Results like these seemed to give strength and reason to the "majoritarian" narrative articulated by Fabricio Alvarado from the very beginning of the dispute and, in fact, constituted a stimulus for his political campaign between January and April. However, what is interesting about the case is that by the second round of elections, held on 1st April, the majority opinions of

the Costa Rican population regarding the resolution by the Inter-American Court either changed direction (from rejection to majority approval) or were insignificant in deciding the final outcome of the election.

[15] For Costa Rican conservative religious activism, Agustín Laje is not a simple intellectual reference but has become a key piece for the ideological articulation of the movement as a whole in recent years. He built close relationships with political parties of evangelical orientation, which invited him to give talks and conferences in the country on several occasions since 2017—a year in which he even gave a conference in the Legislative Assembly (Romero 2017). This reveals that local conservatism participates in transnational intellectual networks and that it seeks to incorporate non-religious ideas into its thinking.

[16] From this perspective, any objection to the historically hegemonic gender and sexual order in Western societies is seen as the result of a global conspiracy that represents a new form of colonialism. In addition, progressive gender and sexuality activism would not only form a transnational movement by virtue of their explicit networks but would also be supported or even led by large capitalist corporations, such as Amazon and Google, international human rights agencies, such as the UN itself, and by billionaires such as George Soros (Graff and Korolczuk 2022). Finally, this global alliance would aim to undertake eugenic control over the growth of populations worldwide.

[17] Local and international conservative activists condense the object of their discomfort and animosity into the notion of "gender ideology". Under this formula, these sectors misinterpret the meaning of scientific theories related to gender and human sexuality and, correlatively, define both the contours of their adversaries and their own political objectives. According to conservative activists, the sectors that subscribe to "gender ideology" have mistaken ideas about human nature and, moreover, seek to impose these "unfounded" ideas on society through indoctrination and the colonization of social institutions (the family, public education, courts and tribunals, etc.) (Paternotte and Kuhar 2017). In Costa Rica, such "ideology" has been systematically denounced in Catholic (Eco Católico) and evangelical (El Camino) media and is often presented as one of the greatest threats to the inherited Judeo-Christian moral order.

[18] In the cases of the PUSC and the PLN—parties that dominated the political-electoral field between the 1980s and the first decade of the 21st century—some factions are considered sympathizers of the global current, but others are not.

[19] Despite the fact that it is impossible to affirm, in the absence of evidence, that the announcement of Advisory Opinion 24/17, in the last stretch of the 2018 electoral campaign, responded to a premeditated strategy of the Costa Rican government in order to obtain political-electoral capital, it is undeniable that this favored the presidential aspirations of the ruling party and its candidate, Carlos Alvarado Quesada. Like his opponent, Fabricio Alvarado, Carlos Alvarado Quesada grew in the polls and managed to access the second round of elections thanks to the position he adopted around what was stipulated by the Inter-American Court. In his case, it was his position of frank support for the Advisory Opinion that finally made him gain the sympathy of an important part of the electorate (Pineda-Sancho 2019).

[20] This was to a certain extent distorted by the forcefulness of the runoff results, where the "progressive" candidate, nemesis par excellence of the conservative option headed by Fabricio Alvarado, obtained a little more than 60% of the total valid votes. This means that the conservative option was defeated in the arena that its own representatives had taken great pains to position before public opinion as the most legitimate space within the democratic dynamics: the ballot box.

[21] For further details, look into Res. N° 2018912782 (Exp: 15-013971-0007-00), Sala Constitucional de la Corte Suprema de Justicia de la República de Costa Rica, San José, 8 August 2018.

[22] From the moment the ruling of the Constitutional House became known, in August 2018, there were conservative legislators who announced their staunch opposition to any law initiative aimed at incorporating the figure of same-sex marriage in Costa Rican legislation (Ruiz 2018). In September 2019, conservative congressmen from parties such as the evangelical Restauración Nacional and the traditional PUSC and PLN promoted a civil union bill for same-sex couples as an alternative to the equal marriage option (Alfaro 2019). This initiative not only aimed to prevent the inclusive expansion of the marriage figure but also sought to deny same-sex couples the right to adoption. Less than a month later, the project collided with the majority opinion of the Legislative Assembly's Human Rights Commission and lost, from that moment on, all political viability (Ramírez 2019).

[23] Faced with the impotent attempts aimed at preventing the inclusive extension of the marriage figure, the conservative legislators tried to delay the inevitable for some months on at least a couple of occasions. In March 2020, the Constitutional Chamber rejected a request for postponement presented by 11 congressmen of the Republic, most of them belonging to the National Restoration faction (Paniagua 2020). In May of that same year, a few days before the deadline stipulated by the Constitutional Chamber in Resolution No. 2018912782, the majority of the deputies of the Republic rejected a couple of motions that sought to delay equal marriage's entry into force for 18 months (Arrieta 2020).

[24] In spite of the narrowly "majoritarian" and "electoralist" conception of democracy that was privileged by conservative religious activism during the controversy analyzed here, at the end of the day it became evident that the democratic regime in force in Costa Rica exceeds the limits of such a definition. As a liberal democracy, the Costa Rican regime is not based exclusively on the will of the majorities, but at the same time contemplates mechanisms to restrict the margin of action of said majorities over the integrity of the minorities and the temporal reproduction of the democratic regime itself. These mechanisms are the Magna Carta, the human rights conventions signed by the state itself in the exercise of its sovereignty, and, of course, the instances in charge of guaranteeing the application of such instruments. Their democratic legitimacy does not derive directly from electoral logic but from the criterion of impartiality aimed at deactivating specific advantages, hoarding, and injustices (Rosanvallon 2020).

25 Three of these initiatives were presented as bills exclusively oriented to the protection of conscientious objection (see legislative files 22.006, 22.263, and 22.785), while the other two incorporated the figure in broader bills (see legislative files 21.012, concerning the "Law for religious freedom and worship", and 10.159, concerning the "Framework Law for public employment"). Most of these initiatives had the approval of some legislators who did not belong to political parties of religious inspiration, which shows, once again, that moral conservatism in Costa Rica is not the exclusive domain of the latter.

26 The initiative that did prosper was the one contained in the "Public Employment Framework Law" (Exp. 10.159). However, in this case, only a very partial form of the figure pursued by conservative activism was adopted. With the approval of this law, public officials were allowed to claim reasons of conscience to be exempted from receiving training programs in matters related to gender equality (and sexual diversity) and human rights. Hence, most of the remaining initiatives are still in the legislative pipeline.

27 A "false right" would be one that is not directly stated in the human rights charters or declarations recognized by the Costa Rican State. While an "authentic right", on the contrary, would be that which is explicitly stated as such in them.

28 The problem with this conservative thesis is that it distorts, either through ignorance or malice, the true argument articulated by those who at the time fought for the legal state recognition of same-sex unions in Costa Rica. According to most of the activists who participated in that movement, marriage as such was not the human right to be claimed but rather unrestricted equality before the law and non-discrimination on sexual grounds. Therefore, the claim was not based on presenting inclusive marriage as a human right but on defending the right of every person and couple to receive the same protections and recognition from the state legal system, which included, of course, access to all the family regulation and protection figures existing in it. This was, in fact, the reasoning followed by the judges of the IACHR when they decided and issued the controversial Advisory Opinion.

29 From the perspective of the political adversaries of conservative activism, it is their adversaries that strategically and self-interestedly select those rights that coincide with their values and political projects. This only reaffirms the existence of a dispute over the legitimate definition of human rights and at the same time evidences the inherently conflictive nature of the political dynamics of pluralist societies, in which the diversity of socio-existential positions, values, and worldviews is more the norm than the exception.

30 With this, the actors that make up conservative religious activism place their values as universal for the whole society and not as attributes circumscribed to particular sectors of it. "The others" speak from the perspective of the universal, of the human par excellence, while the "Others" speak from a particular interest, deforming the true essence and purpose of humanity.

31 Although it will not be analyzed in this article, it should be noted that the notion of "natural law" is an essential pillar in the political positioning of conservative religious activism. They generally articulate their thinking on issues such as democracy, human rights, and the legal system (or positive law). At the same time, they use it as a basis for criticizing relativist, constructivist, and sociological trends of thought, which they associate with the destruction of the foundations of the human being and the order designed by the Creator. The use of this notion allows conservative actors to conceal the theological-religious underpinning of their public positions, since it is "easily" associated with biological criteria (scientific, naturalistic, etc.), but at the same time offers them the possibility of remaining anchored in it.

32 Based on the analysis of 60 interviews conducted during 2018 regarding the electoral situation of interest, Molina and Tretti (2021) found that, among all the candidates who participated in the process, Fabricio Alvarado was the one who tended to receive a greater number of negative evaluations regarding his potential capacity to assume the presidency and exercise it adequately. While the ruling party candidate, Carlos Alvarado, the eventual winner of the race, was the one who, comparatively, received more positive evaluations in this area.

33 According to a survey conducted by the Center for Political Studies of the UCR after the second round, the three factors that most influenced the vote for Carlos Alvarado of the PAC were his performance during the campaign, the defense of the Costa Rican rule of law, and the safeguarding of patriotic values (CIEP 2018b, p. 25). In one way or another, those who voted for the candidate of the ruling party perceived the potential triumph of the evangelical Fabricio Alvarado as a threat both to the normative framework of the Costa Rican political regime (including its guarantees and freedoms) and its cultural background. For its part, the lack of experience that many of these people attributed to him, with or without justification, and the fear that his belligerent and incendiary speech arose, among many others, played against the evangelical candidate.

34 The organizational weakness and lack of cohesion of the PRN became even more evident during the months immediately following the runoff election. As soon as its representatives in the Legislative Assembly began to take office, internal quarrels became the tone of the administration. Differences between the old and new leaderships within the party led Fabricio Alvarado and his closest supporters to split from the political party and form a new one: the Partido Nueva República (Rodríguez 2018). In practice, this early split not only weakened the PRN within Congress but also debilitated evangelical action within the Costa Rican political arena. The PRN was unable to consolidate itself as a compact bloc.

35 Not even the constitutional status of the Catholic Church in Costa Rica has prevented such progress. Despite having a low index of formal secularism, the country currently presents "a high score in the fulfillment of sexual and reproductive rights" (Católicas por el Derecho a Decidir 2020, p. 61). This situation relativizes the socio-political weight of state confessionality and allows us to rethink, consequently, the categories through which secularity and its practical (concrete, everyday) effects on social life are understood and measured at a theoretical and empirical level.

36     This makes the phenomenon analyzed here fertile ground for the application of a history and a conceptual sociology of the political. This sub-discipline, or line of study, has been developed and practiced by scholars such as Pierre Rosanvallon in Europe and José Elías Paltí, in Latin America. This approach seeks "to understand the formation and evolution of the representation systems [or political rationalities] that govern the way in which an era, a country or social groups conduct their actions and imagine their future" (Rosanvallon 2002, p. 128), as well as to identify the conflicts and intergroup controversies that take place in them.

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
