# Peer review of "The Dispute around Same-Sex Marriage in Costa Rica: Arguments and Actions of Conservative Religious Activism (2017–2021)"

_religions, doi:10.3390/rel14040540_

Round 1

Reviewer 1 Report

This article is a narrative analysis of a recent event in the politics of Costa Rica. The author examines the controversy about the legalization of same-sex marriage, focusing on the activism of religious conservatives during the presidential election of 2018. Conservative religion has long opposed changes in legal structures supporting traditional values regarding issues of gender and sexuality, but it is important to be attentive to particularities in processes with regard to place and time.  This article does both, bringing attention to the Costa Rican context.  The strength of this article is its analysis of a recent change in the positioning of the religious right, which now incorporates specific claims about democracy and human rights into their arguments.

The author succeeds in providing enough details about the events under consideration that a reader unfamiliar with Costa Rican politics can follow the argument, without being overwhelmed in the data.  The writing is clear, and the narrative well presented. Shortly before the election, the Costa Rican government made public an Advisory Opinion of The Inter American Court of Human Rights (IACHR) urging Costa Rica to recognize and protect the relationships of all individuals, granting same sex couple the same rights and privileges as heterosexual couple.  The religious conservatives mobilized against the ruling, but the rhetoric they used as secular, arguing for the protection of their human rights, and the preservation of democracy.  They argued that the Costa Rican government acted as part of a conspiracy with the IACHR to undermine democracy and disregarded the views of the (fictious) majority of Costa Ricans. Furthermore, their rhetoric positioned the religious conservatives as the true preservers of democracy in the face of a colonizing world order (George Soros, Amazon) that seeks to erode the autonomy of all nations.  The author shows how the rhetoric in the 2018 election with its claims about how human rights are defined is a departure from appeals to the word of God or church doctrines.  The author makes the important point that the religious groups are adapting and reacting the political grammar of the times.  The author also argues for a growing trend in Costa Rica toward moral pluralism, with movement toward recognizing Sexual and reproductive Health Rights gaining ground at legal, institutional and cultural levels.  The author also uses this story to reflect on secularism in Costa Rica, the only country in the Americas with a state church.  Although the 1949 constitution guarantees freedom of religion, it also retained the Roman Catholic church as the official state religion. In this election Catholics and Evangelicals built an alliance of religious conservatives. That religious actors have entered the political debates using the language of human rights and democracy left them struggling about definitions about terms in a political language they do not control is informative about the dilemmas the conservatives face. 

The topic is timely and valuable. The Costa Rican case presents an interesting example of conservative religious mobilization in a context of increasing moral pluralism. 

It would be useful to hear more about intervention of the IACHR.  What motivated it?  How does the author interpret the timing of the release by the government of the Order?  How did ordinary Costa Ricans respond?

Author Response

Point 1: It would be useful to hear more about intervention of the IACHR.  What motivated it?  How does the author interpret the timing of the release by the government of the Order?  How did ordinary Costa Ricans respond?

Response 1:

(Footnote p. 1)

This advisory opinion was expressly requested by the Costa Rican state on May 18, 2016 (IACHR, 2017, p. 3). In this case, the Costa Rican government exercised its right to consultation granted by Article 64 of the American Convention on Human Rights, in order to clarify whether the San Jose Pact protected the recognition of name changes for individuals based on their self-perceived gender identity, and if the American Convention recognized property rights derived from relationships between same-sex couples (Presidencia de la República, 2016). According to the promoters of the initiative, the Costa Rican government sought to resolve by this means the historical lack of state recognition (legislative, judicial, etc.) of the demands and rights of sexually diverse citizens (González, 2020).

(Footnote p. 2)

The Inter-American Court of Human Rights adopted the Advisory Opinion discussed here on November 24, 2017 (CIDH, 2017), but the Costa Rican government did not receive it until January 9, 2018 (Presidencia de la República, 2018); day in which the international court issued a press release on the matter (CIDH, 2018). Therefore, the delay in the announcement of Advisory Opinion 24/17 is not attributable to a premeditated strategy of the government of Costa Rica in order to obtain electoral capital, as its detractors and political adversaries stated at some point in the dispute, but rather responded to the dynamics of the Inter-American Court. That is all that can be stated, on a non-speculative basis, from the available sources.

(Footnote p.19)

Despite the fact that it is impossible to affirm, in the absence of evidence, that the announcement of Advisory Opinion 24/17, in the last stretch of the 2018 electoral campaign, responded to a premeditated strategy of the Costa Rican government in order to obtaining political-electoral capital, it is undeniable that this favored the presidential aspirations of the ruling party and its candidate, Carlos Alvarado Quesada. Like his opponent, Fabricio Alvarado, Carlos Alvarado Quesada grew in the polls and managed to access the second round of elections thanks to the position he adopted around what was stipulated by the Inter-American Court. In his case, it was his position of frank support for the Advisory Opinion that finally made him gain the sympathy of an important part of the electorate (Pineda-Sancho, 2019).

(Footnote p.14)

According to a public opinion survey conducted by the Center for Research and Political Studies of the University of Costa Rica between January 15th and 17th, 2018, one week after the announcement of the Advisory Opinion, 75% of the Costa Rican population was aware of the resolution at that time, and among this percentage, 59% of people expressed opposition to its contents (CIEP, 2018a, p. 8).

Results like these seemed to give strength and reason to the "majoritarian" narrative articulated by Fabricio Alvarado from the very beginning of the dispute and, in fact, constituted a stimulus for his political campaign between January and April. However, what is interesting about the case is that by the second round of elections, held on April 1st, the majority opinions of the Costa Rican population regarding the resolution by the Inter-American Court either changed direction (from rejection to majority approval) or were insignificant in deciding the final outcome of the election.

Reviewer 2 Report

The article focuses on a very current issue in Latin America, such as the relationship of conservative religious movements with partisan politics, in resistance to the incorporation of new rights such as same-sex marriage.
The article is well organized and explores the topic in depth, focusing on a very particular historical moment in Costa Rica.
The analysis of the link between conservative religious groups and democracy is very interesting and it considers the concrete way in which this link exists in Costa Rica.
The only weakness I find in the work is that it takes for granted many issues of the Costa Rican reality that may limit the full understanding of it by people who have no knowledge of that reality.
I am referring to what the text calls "Costa Rican conservative religious activism" which does not mention who is included in this category.
It mentions "conservative religious activism with political-electoral representation".  The existence of confessional parties is a relevant fact, so I understand that it should be clarified what is being referred to.
At another point it mentions the following: "parties of evangelical orientation with representation in Congress since 1998". Thus, it is implied that there have been religious parties with representation in Congress for quite some time, but at no time does it make reference to which ones they are, nor what positions they hold.
Has the resistance to the same-sex marriage law been only from the parties with representation since 1998 or did it also include other parties or other groups?
Having said this, I consider that, in order to be published, the article should include a description of the assumptions used by the author in the mentioned matters.

Author Response

Point 1: The only weakness I find in the work is that it takes for granted many issues of the Costa Rican reality that may limit the full understanding of it by people who have no knowledge of that reality.

I am referring to what the text calls "Costa Rican conservative religious activism" which does not mention who is included in this category.

It mentions "conservative religious activism with political-electoral representation".  The existence of confessional parties is a relevant fact, so I understand that it should be clarified what is being referred to.

At another point it mentions the following: "parties of evangelical orientation with representation in Congress since 1998". Thus, it is implied that there have been religious parties with representation in Congress for quite some time, but at no time does it make reference to which ones they are, nor what positions they hold.

Has the resistance to the same-sex marriage law been only from the parties with representation since 1998 or did it also include other parties or other groups?

Having said this, I consider that, in order to be published, the article should include a description of the assumptions used by the author in the mentioned matters.

Response 1:

(footnote p. 3)

This category includes civil society organizations such as the Conferencia Episcopal de Costa Rica (CECOR) y la Federación Alianza Evangélica Costarricense (FAEC), political society actors such as the Renovación Costarricense (PRC) y Restauración Nacional (PRN), both political parties with evangelical orientation, and the Costa Rican Catholic Church as the official church of the Costa Rican state. These were the institutions that were most actively and visibly involved in the controversy discussed in this article.

However, it is necessary to highlight that in practice, it was the Partido Restauración Nacional (PRN), led by Fabricio Alvarado Muñoz, the conservative actor that had the most prominence rol in the dispute and the one that was at the forefront of the conservative cause. This happened in such a way by virtue of the electoral nature that the controversy acquired and the ability that the PRN had to put it at the service of its own political ends.

(footnote p. 7)

Conservative religious activism with political-electoral representation includes, in a restricted sense, political groups with an explicitly religious orientation that have direct participation in the political-electoral field (political society) and presence in the Congress of the Republic. For the case discussed in this article, reference is made, specifically, to the evangelical parties Renovación Costarricense (PRC) and Restauración Nacional (PRN).

(footnote p. 12)

In Costa Rica, there have been evangelical political parties since 1981. However, it was not until 1998 that they managed to obtain representation in the Congress of the Republic. Since then, these groups have been able to maintain a minority presence in Congress and a limited share of power within the electoral political field. In both realms, they have been champions of the conservative cause in favor of maintaining the moral status quo in Costa Rica (Pineda-Sancho, 2019; 2023).

(footnote p. 8)

The opposition to same-sex marriage was not, as has been pointed out in different parts of this article, an exclusive stance of religious parties with representation in Congress. This position was also taken, during the election campaign, by members of non-religious (or at least, not directly religious) political parties and by secular actors in civil society. If the paper doesn’t give enough space to those actors, it is because their role in the controversy was not particularly prominent and because the political project of these actors is not primarily based on religious concerns and ideals.